# An Investigation into the Effect of Electro-Contact Heating in the Machining of Low-Rigidity Thin-Walled Micro-Machine Parts

**DOI:** 10.3390/ma14164427

**Published:** 2021-08-07

**Authors:** Antoni Świć, Arkadiusz Gola, Olga Orynycz, Karol Tucki

**Affiliations:** 1Department of Production Computerisation and Robotisation, Faculty of Mechanical Engineering, Lublin University of Technology, ul. Nadbystrzycka 36, 20-618 Lublin, Poland; a.swic@pollub.pl; 2Department of Production Management, Bialystok University of Technology, ul. Wiejska 45A, 15-351 Bialystok, Poland; o.orynycz@pb.edu.pl; 3Department of Production Engineering, Institute of Mechanical Engineering, Warsaw University of Life Sciences, ul. Nowoursynowska 164, 02-787 Warsaw, Poland; karol_tucki@sggw.edu.pl

**Keywords:** turning, thin-walled parts, low-rigidity parts, micro-machines, electro-contact heating

## Abstract

Low-rigidity thin-walled parts are components of many machines and devices, including high precision electric micro-machines used in control and tracking systems. Unfortunately, traditional machining methods used for machining such types of parts cause a significant reduction in efficiency and in many cases do not allow obtaining the required accuracy parameters. Moreover, they also fail to meet modern automation requirements and are uneconomical and inefficient. Therefore, the aim of provided studies was to investigate the dependency of cutting forces on cutting parameters and flank wear, as well as changes in cutting forces induced by changes in heating current density and machining parameters during the turning of thin-walled parts. The tests were carried out on a specially designed and constructed turning test stand for measuring cutting forces and temperature at specific cutting speed, feed rate, and depth of cut values. As part of the experiments, the effect of cutting parameters and flank wear on cutting forces, and the effect of heating current density and turning parameters on changes in cutting forces were analyzed. Moreover, the effect of cutting parameters (depth of cut, feed rate, and cutting speed) on temperature has been determined. Additionally, a system for controlling electro-contact heating and investigated the relationship between changes in cutting forces and machining time in the operations of turning micro-machine casings with and without the use of the control system was developed. The obtained results show that the application of an electro-contact heating control system allows to machine conical parts and semi-finished products at lower cutting forces and it leads to an increase in the deformation of the thin-walled casings caused by runout of the workpiece.

## 1. Introduction

Axially symmetrical parts make up about 34% of the total production in the machine industry, 12% of which are low-rigidity shafts [1]. Parts of this type are used in the aviation industry [2], precision mechanics [3], the tool industry (special tools) [4], and the automotive industry [5]. They have irregular shapes and low stiffness in specific cross-sections and directions [6]. During production, high requirements are placed on their geometric parameters, mutual position of the surfaces, linear dimensions, and surface quality [7,8,9].

Among the large variety of low-rigidity rotating parts, a special place belongs to thin-walled parts, which are characterized by specific stiffness-to-weight ratios [10]. The dimensions of thin-walled parts are expressed by the dimensionless coefficients *β_t_* = *H*/*D*
*>* 15 and *α_t_* = *d*/*D* ≤ 0.9 (where *H*, *D*, and *d* stand for the height and external and internal diameters of cylindrical surfaces, respectively) [11]. Although machining of this type of parts is important from the point of view of the current market needs, the literature does not offer much information on the control of elastic displacements during machining of thin-walled parts, and the existing studies mainly go as far as describing various tools and devices using steady rests [12], tool holders, frictionally damped rotary boring bars [13], multi-tool holders or devices for double-sided machining, which provide additional support and only help to reduce the deformation of the part, and not to control elastic deformation [14].

When machining low-rigidity thin-walled rotating parts, it is very difficult to achieve the required accuracy of shape and dimensions, and surface quality [15,16]. The low inherent stiffness and the relatively low rigidity of these parts in comparison to the rigid machine tool assemblies, lead to the formation of vibrations under certain conditions, which significantly impede the control of the process [17]. Moreover, machining is affected by numerous interfering and destabilizing factors (large residual strains, the machine tool, the fixture, swarf, dust, etc.), which compromise the quality of the parts produced [18]. As a consequence, the desire to obtain appropriate surface quality parameters while ensuring a high machining performance spurs the search for new methods of producing thin-walled parts, using machining parameters similar to those used in the production of normal-rigidity parts [19,20,21].

Unfortunately, traditional machining methods which use multi-pass machining, reduced cutting parameters, or additional treatments, cause a significant reduction in efficiency, and in many cases do not allow to obtain the required accuracy parameters [22]. Moreover, they also fail to meet modern automation requirements and are uneconomical and inefficient. Consequently, it is necessary to search for new methods that will enable high-efficiency production of this type of part with specific quality parameters.

The aim of this study was to identify the dependency of cutting forces on cutting parameters and flank wear, as well as changes in cutting forces induced by changes in heating current density and machining parameters during the turning of thin-walled parts. To reach the defined goal the test stand based on an SNA 400/1000 lathe was designed and a system for controlling electro-contact heating was developed. These allowed to determine changes in cutting forces and temperature depending on cutting speed, feed rate, and depth of cut, and a block diagram of a specially designed control system for electro-contact heating of the cutting zone in the operation of machining micro-machine casings.

The article comprises four sections. Section 1 presents the theoretical aspects of the dynamic character of machining of low-rigidity thin-walled parts, the importance, challenges, and the necessity of developing proper control methods and models for automation of such processes, and a review of the relevant literature. Section 2 describes the characteristics of the machined parts and a test stand used for provided experiments. Moreover, characteristics of the developed electro-contact heating control system were provided in this part of the paper. Section 3 reports the results obtained during the experimental studies and necessary discussion. In particular, this section was focused on the investigation of the effect of machining type and machining parameters on the cutting operation (Section 3.1) and presents the results of the use of the electro-contact heating control system (Section 3.2). The article concludes with Section 4, which contains observations and reflections made during the experiments, analyses, and modeling.

## 2. Materials and Methods

### 2.1. Characterization of the Machined Parts

The types of low-rigidity parts analyzed in this study are components of many machines and devices, including high precision electric micro-machines used in control and tracking systems.

The accuracy class of electrical devices can be increased by using high-quality materials that ensure minimum distortion of electro-magnetic characteristics. For this reason, thin-walled casings of electric micro-machines are produced from Grade 2 and Ti-64 titanium alloys [23,24]. Increased demands are also placed on the accuracy of the manufacture of similar parts and the stability of the dimensions of a part a long time after it has been machined [25]. In this connection, the technological problems of ensuring the dimensional stability of high precision parts and the automation of the process of machining those parts still remain topical.

Semi-finished casings for electric micro-machines (Figure 1) are made by sintering Ti-64 titanium powder. The possibility of such elements getting deformed in the interoperational cycle depends on the presence of internal stresses in them, the highest of which occur during the manufacture of semi-finished products and roughing. Semi-finished parts are characterized by different hardness, ranging from 73 to 110 HB, already after the titanium powder has been pressed and sintered [26]. This is due to the different oxygen content in the titanium alloy casings, as well as the different density of the structure of the semi-finished products (semi-finished parts produced from low-dispersion powder have a higher hardness and density) [27].

The substantial technological taper (conicity) of the semi-finished casings (Figure 1) is a consequence of the pressing process. The machining operation contributes to the formation of an asymmetrical field of ultimate internal stresses in the axial direction and along the circular coordinate, which leads to instability of the dimensions of the casing over time and exerts an effect at the interface between the micro machine casing and magnetic wires. This in turn results in a sharp reduction in the precision of the electric micro-machine. The machining process is accompanied by intensive wear of the cutting edge, which also affects the quality and dimensional stability of the thin-walled precision part [28].

### 2.2. Schematic of the Test Stand

A test stand was constructed using an SNA 400/1000 lathe to determine the cutting forces *F* and temperature *θ* at different values of cutting speed *v_c_*, feed rate *f*, and depth of cut *a_p_*. The test stand is shown in Figure 2. The device included a workholding device (a mandrel) 1 holding a workpiece 2, fitted in a dielectric sleeve 3, and the machine tool holder 4. The chuck was pressed by the rear center 5 of the machine tool, mounted in the dielectric sleeve 6 of the tailstock 7. The cutting edge 8 was positioned in the tool holder 9 of a tool post 10. The temperature in the cutting zone was measured using the “tool–workpiece” natural thermocouple technique with a ring 11 fitted with a current measuring device 12, a vacuum switch 13 with an electrodynamic drive (the switching time from the cutting zone heating mode to the cutting temperature control mode did not exceed 0.025 s), and a digital F4214 voltmeter 15 (measurement error ±0.1 μV).

Components of the cutting force were determined using a two-component dynamometer 17 (base length of sensors 30 mm, active resistance 100 ± 0.2 Ω), a UT4-1 type tensostation, C4311-type devices 20, and an N327/3-type recording device 19. The dynamometer 17 was tarred using a standard DOSM-0,2 spring dynamometer. Power was supplied to the measuring setup and the test stand using R-1000 type ferroresonance voltage stabilizers. The workpiece 1 was electro-contact heated to a tempering temperature of the surface layer using an EKR-14 device, which consisted of a PNO 5-520 transformer, a transformer with a falling characteristic, rectifiers, and a capacitor for extinguishing arcs when introducing and removing the cutting edge into and out of the cutting zone. A direct current was used because we wanted to check its effectiveness when stabilizing and not stabilizing the technological ultimate stress field.

Cemented carbide H30 was selected as the material for the working part of the cutting tool used in the process of roughing with electro-contact heating. This material is characterized by minimal electrical resistance *ρ* = 18.6 · 10^−8^ Ω compared with other materials in this group. Increasing the temperature leads to a drop in *ρ* for cemented carbide H30 and a slight increase in specific electrical resistance for Grade 2 and Ti-64 alloys, which guarantees better heating of a workpiece with higher electrical resistance when the “tool-workpiece” chain is arranged serially.

Flank wear was measured using an MMI-8 microscope (geometric parameters of turning: rake angle *γ* = 0°, clearance angle *α* = 10°, cutting edge angle *κ_r_* = 75°, end cutting edge angle *κ_p_’* = 15°, inclination angle of side cutting edge *λ_s_* = 0°, nose radius *r_p_* = 1.0 mm; boring parameters: *γ* = 0°, *α* = 12°, *κ_r_* = 94°, *κ_r_’* = 10°, *λ_s_* = 0°, *r_r_* = 1.0 mm; cutting parameters *v_c_* = 1.3 m/s, *f* = 0.15 mm/rev, *a_p_* = 1.5 mm). A constant speed in the range of 4–5% was maintained owing to selecting an appropriate workpiece diameter and switching the electric clutches of the machine tool gearbox. Precise measurements of the rotational speed of the spindle were made using an ST-5 strobe tachometer.

### 2.3. Schematic of the Electro-Contact Heating Control System

A block diagram of the system for the control of electro-contact heating in the cutting zone during the machining of micro-machine casings is shown in Figure 3.

The adaptation device was constructed on the basis of an electro-contact heating program control system and comprised a cutting force sensor 15, a threshold element 16, and a program block 17.

After the cutting edge had cut into the workpiece, when the cutting force reached an appropriate value, the signal from the sensor 15 was transferred to the threshold element 16 set at an appropriate start-up voltage. The signal from the threshold element 16 activated the electro-contact heating device 14, and the electric current flowed through the current-feed ring 11 and contacts 12–13, to the workpiece 1. By adjusting the heating current density, the cutting forces could be stabilized along the length of the pass. At the end of the pass, after the cutting tool 8 had been withdrawn, the threshold element 16 disengaged the device 14, preventing in this way the formation of an arc. Because the roughing pass reduced the cross-sectional area of the layer, the cutting force value stored in program block 17 for each of the next passes was decreased.

## 3. Results and Discussion

### 3.1. Effect of Machining Type and Machining Parameters on the Cutting Operation

#### 3.1.1. Effect of Cutting Forces

To obtain the required efficiency of machining of parts made from Grade 2 and Ti-64 alloys, the machining allowance must be removed in one pass [29]. An analysis of experimental data (Table 1) shows that the cutting forces generated during turning and boring increase from 2.6 to 3.7 times, with significant pulsation at the beginning of machining. The significant increase in cutting forces can be explained by an increase in the depth of cut from *a_p_* = 1 mm to *a_p_* = 2.7 mm, and the pulsation of the cutting forces in the initial stage of machining is due to the fact that workpiece runout and depth of cut have similar values.

An analysis of the data in Table 2 shows that increasing the temperature improves the machinability of *Grade 2* and *Ti-64* alloys as their strength characteristics decrease and the shape stability coefficient *K_k_* significantly increases.

Curves of cutting forces vs. cutting speed in boring and turning operations are shown in Figure 4 and Figure 5.

An analysis of the curves in Figure 4 shows that the main cutting forces *F_c_* and thrust cutting force *F_p_* are 25–27% and 22–25% lower, respectively, when the material being machined is metal-ceramic titanium compared to uniform titanium. It should also be noted that *F_c_ ≈ F_p_* when *Ti-64* workpieces are subjected to boring at the cutting edge angle *κ_r_* = 94°. This is very important when shaping the field of technological ultimate stresses because in this case, the effect of cutting tool flank surface on the workpiece is clearly visible. Chip formation is more difficult, which is expressed in higher values of the cutting edge angle compared to external turning.

During external turning, the decrease in *F_c_ = f (v)* at *f =* const and *a_p_ =* const (Figure 5) is greater in the case of *Ti-64* than *Grade 2*, which is a consequence of a more intense increase in temperature in the cutting zone (other conditions being comparable) due to the reduced thermal conductivity of metal-ceramic titanium compared to uniform titanium.

Thin-walled casing workpieces subjected to roughing bear traces of sintering in the form of nitrates and titanium oxides. There is significant wear of the cutting edge caused by the action of titanium nitrate and thermally-induced adhesive wear. The results of the experiments testing the effect of flank wear *h_p_* on the value of the cutting force components *F_c_* and *F_p_* in the operation of boring metallo-ceramic titanium are shown in Figure 5b. An analysis of this relationship shows that in the operation of boring *Ti-64*, *F_p_* increases more intensively than *F_c_*, and the curve has practically no horizontal section; at *h_p_* = 0.35 × 10^−3^ m, which corresponds to the maximum permissible flank wear, the cutting forces increase from 1.5- to 1.7-fold. Such a sharp increase in cutting forces explains the effect of wear rate *h_p_* on the long-term dimensional stability of thin-walled precision parts, because, as the cutting forces grow, so does the intensity of the technological ultimate tensile stress field, the relaxation of which leads to large deformations of the thin-walled sleeves.

#### 3.1.2. Effect of Cutting Parameters on Temperature Growth

It is known that an increase in temperature leads to an increase in the machinability of technical titanium and metal-ceramic titanium, as the strength properties of these materials rapidly decrease. In this connection, we conducted tests to determine the effect of cutting parameters on the cutting temperature. An analysis of this relationship (Figure 6a) shows that the depth of cut at *f* = const has practically no effect on the machining temperature. The elevating feed rate f from 0.12 to 0.212 mm/rev leads to an increase in temperature by 35–42 °C (Figure 6b). Changes in the cutting speed (Figure 6c) in the range from 1.3 to 3.0 m/s at *f* = const and *a_p_* = const lead to an increase in the average cutting zone temperature by 15–18 °C.

A change in the cutting speed during the machining of Ti-64 workpieces caused by their conicity is 4.5–5%. According to the experimental data, the cutting temperature does not change by more than 4.5–6 °C during the machining of one part. This means that changes in cutting speed can be ignored as a factor influencing the machining process. It was assumed that *F* = const, *v* = const and *a_p_* = var; because *F_n_* = *f* · *a_p_* ∙ cos κ, where *F_n_* is the cross-sectional area of the machined layer during 1 rotation of the workpiece, with the proviso that *a_p_ = f(L)* and the cutting temperature when using electro-contact heating is directly proportional to electric current density squared, the optimal current density *I_g.op_* will correspond to a specific temperature *θ°_opt_*. A/m^2^ at specific *v_c_* and *f*, which can be maintained by increasing the current fed to the cutting zone proportionally to increasing the depth of cut along the machining length.

#### 3.1.3. Effect of Heating Current Density and Turning Parameters on Changes in Cutting Forces

The dependence of flank wear on the current density for the cutting parameters corresponding to rough turning (boring) of *T-64* and *Grade 2* alloys is shown in Figure 6d. Increasing the cutting speed leads to a certain reduction in *I_g_*_.*op*._ (curves 3, 4, Figure 6d), but at the same time reduces the durability of the parts produced, as the degree of the impact of abrasive inclusions and the intensity of micro-impacts becomes larger. An analysis of the curves shows that when the “tool–workpiece” setup is used, the optimum value of current for workpieces made of *Ti-64* according to the criterion of tool flank life is *I_g_*_.*op*._ = 400 A/(m^2^ · 10^−6^) for external turning—curve 1, and *I_g_*_.*op*._ = 470 A/(m^2^ · 10^−6^) for boring—curve 2.

At the same time, tool life increases. Compared to *Grade 2*, this is a small increase, which can be explained by the effect of *TiN* particles at grain boundaries and intense micro-impacts. In the process of machining *Grade 2*, tool life increases from 3.5 to 5 times. The lower value of the optimal current density for the boring operation is due to the higher cutting temperature used during boring compared to external turning, other conditions being equal.

To check whether the dimensional stability and long-term dimensional stability of thin-walled casings could be enhanced using electro-contact heating of the cutting zone at a constant optimal current density along the machining length and to amend the mathematical model of the process model so that it took into account the reduction in cutting forces at a constant optimal current density along the length of the workpiece, we designed and conducted 2^4^ experiments.

An analysis of Figure 7a, plotted based on experimental data (external turning), shows that the main cutting force *F_c_* for *I_g_*_.*opt*._ = 470 A/(m^2^ · 10^−6^) and, accordingly, *θ*°*_skr_* = 900 °C decreases by 18–22%, and the feed component *F_f_* drops by 13–15%. A further increase in current density leads to an intensification of flank wear (Figure 6d) and an increase in the cutting forces and the temperature in the cutting zone.

The last two factors increase the slope of the flank wear growth curve, which at the same time leads to an increase in the deformation of the thin-walled casings. The smaller decrease in *F_f_* is due to the high value of the cutting edge angle (*κ_r_* = 75°), which is conditioned by the tendency/which was used to reduce the radial component *F_p_*. Changes in different cutting parameters affect the force relationships in different ways. Increasing the feed rate leads to a change in the slope of *F_f_ =* (*f*, *I_g_*), at *v_c_* = const and *a_p_* = const, causing practically no decrease in *F_c_ =* (*f*, *I_g_*) (Figure 7b). The large influence of feed rate on the slope of the *F_f_* curve is a consequence of the more intense densification and deformation of the metal-ceramic composite in the axial direction, while the intensity of densification in the tangential direction is constant due to the invariability of cutting speed *v_c_*. The densification of the metal-ceramic composite structure undoubtedly leads to a decrease in the specific electrical resistance of the workpiece in the axial direction, which causes a certain increase in the heat flux in this direction. Increasing the depth of cut *a_p_* leads to practically no decrease in *F_f, c_ = f* (*f*, *I_g_*) at *v_c_* = const and *f* = const, (Figure 7c), which is due to the weak effect of *a_p_* on the cutting temperature. In the operation of *Ti-64* boring, *F_c_* and *F_p_* also decrease (Figure 7d), with the radial component *F_p_* decreasing more intensively at *I_g.op._* = 400 A/(m^2^·10^−6^); *F_c_* decreases from 17 to 18% and *F_p_* from 20 to 25%.

### 3.2. Use of the Electro-Contact Heating Control System

Curves of changes in cutting forces along the machining length of micro-machine casings obtained in tests with and without the use of the electro-contact heating control system are shown in Figure 8. When electro-contact heating is controlled, the cutting forces increase more monotonously, they stabilize earlier, and their values decrease. This means that the use of the electro-contact heating control system enables machining of conical parts and workpieces at lower cutting forces while increasing tool life and the quality parameters of the machined parts.

A graphic interpretation of the test of the long-term dimensional stability of the casings in three sections is presented in Figure 9. The dimensions of rough-machined and thermally stabilized thin-walled *Ti-64* micro-machine casings change after 6 days; however, the time of dimension stabilization is not the same for operations using vs. not using electro-contact heating. The use of electro-contact heating reduces the time of dimension stabilization to 24–36 h, which is associated with the lower impact of the cutting process and thermal stability. These conditions favor the formation of a weaker technological ultimate stress field compared to treatment without the use of electro-contact heating.

A comparative analysis of the curves (Figure 9a,b) shows that runout *δ* of the workpiece affects the deformation of the thin-walled casing over time, but does so to a lesser extent when the electro-contact heating control system is used. The maximum increase in dimensional stability in the range from 22 to 27% is observed in section *II* of the part with the lowest stiffness. An increased workpiece runout (Figure 9a) slows down the process of stabilizing the dimensions of the machined part, and parts with minimum runout (*δ* = 0.2 mm—Figure 9b) stabilize faster. The use of the electro-contact heating control system does not change the nature of the time curve of changes in the dimensions of the thin-walled *Ti-64* casings but reduces their absolute value.

An analysis of the curves presented in Figure 9b and c shows that the dimensions of parts with a higher hardness (*HB* ≥ 88—fine-grained *Ti-64*fraction), machined using electro-contact heating of the cutting zone, stabilize to a greater extent compared to soft parts (*HB* ≤ 88—coarse-grained *Ti-64* fraction). This is due to the fact that in the case of soft parts, which have a coarser-grained structure, the ultimate stresses accumulated in individual *Ti-64* grains deform the material in the direction of least resistance, i.e., towards the micro-voids in the workpiece. In the hard parts, the micro-voids have a much smaller volume because work hardening of the fine-grained fraction is not higher than 6% and work hardening of the coarse-grained fraction can even be as high as 15%. This gives rise to the higher dimensional stability of the coarse-grained casings. Greater stabilization of dimensions is obtained with the use of the electro-contact heating control system for casings of higher hardness because their fine-grained, compressed *Ti-64* fraction has better thermal conductivity. When the “tool—workpiece” electro-contact heating control system is used, the fine-grained fraction heats up stronger and more evenly, which enables a greater reduction in cutting forces compared to the machining of coarse-grained workpieces.

## 4. Conclusions

The article presents a test stand based on an SNA 400/1000 lathe designed to determine changes in cutting forces and temperature depending on cutting speed, feed rate, and depth of cut, and a block diagram of a specially designed control system for electro-contact heating of the cutting zone in the operation of machining micro-machine casings. The experiments lead to the following conclusions:
Cutting forces:
-cutting forces generated during turning and boring increase from 2.6 to 3.7 times with considerable pulsation at the beginning of machining. This is caused by an increase in the depth of cut, and the pulsation of cutting forces in the initial stage of machining is a function of workpiece runout and the depth of cut; on the other hand, increasing the temperature improves the machinability of the materials as it leads to a reduction in their strength characteristics and a significant increase in the shape stability coefficient; increasing the temperature improves the machinability of *Grade 2* and *Ti-64*, as their strength characteristics decrease, and significantly increases the shape stability coefficient;-the main cutting forces and the thrust cutting force generated during the machining of metal-ceramic titanium are from 25 to 27% and from 22 to 25% lower, respectively, than for uniform titanium;-the cutting forces during the boring of *Ti-64* increase from 1.5 to 1.7 times, which leads to an increase in the intensity of the technological field of ultimate stresses, the relaxation of which leads to large deformations of the thin-walled sleeves.
Effect of cutting parameters on temperature growth:
-depth of cut at *f = const* has practically no effect on the machining temperature;-increasing feed rate *f* from 0.12 to 0.212 mm/rev leads to an increase in temperature by 35–420 °C;-changes in cutting speed (at *f* = *const*, *a**_p_* = *const*) in the range from 1.3 to 3.0 m/s cause an increase in the average cutting zone temperature by 15–18 °C.
Effect of current density on changes in cutting forces generated during the turning of *Grade 2*:
-main cutting force *F_c_,* at *I**_g.opt_* = 470 A/(m^2^·10^−6^) and *θ*°*_skr_* = 900 °C, declines by 18–22%;-feed component *f* decreases in the range from 13 to 15%;-a further increase in current density leads to an intensification of flank wear and an increase in the cutting forces and the temperature in the cutting zone.
Effect of cutting parameters on changes in cutting forces in the operation of turning *Ti-64*:
-increasing the feed rate leads to a change in the slope of the *Ff* = (*f*, *I*_g_) curve, at *v_c_* = const, *a_p_* = const, causing practically no decrease in the *F_c_* = (*f*, *I*_g_) curve;-increasing the depth of cut *a**_p_* causes practically no decrease in the *F_f c_* = (*f, I_g_*) curve, at *v_c_* = const, *f* = const, which is associated with the weak effect of *a**_p_* on cutting temperature.
The application of the electro-contact heating control system allows machining conical parts and semi-finished products at lower cutting forces. It also leads to an increase in the deformation of the thin-walled casings caused by the runout of the workpiece. Additionally, it was found that the dimensions of harder parts stabilize better (compared to parts with a lower hardness).


## 5. Patents

Lathe tailstock [patent no. 212961]/Victor Taranenko, Antoni Świć, Dariusz Wołos, Georgiy Taranenko. Victor Taranenko, Antoni Świć, Dariusz Wołos, Georgiy Taranenko.-Patent no.; Patent application no.//Official Gazette of the Patent Office, 2012, No. 12, p. 2897.Lathe tailstock [patent no. 211537]/Victor Taranenko, Antoni Świć, Dariusz Wołos, Georgiy Taranenko; author: Victor Taranenko, Antoni Świć, Dariusz Wołos, Georgiy Taranenko.-Patent no.; Patent application no.//Official Gazette of the Patent Office, 2012, No. 5, p. 1073.Lathe tailstock [patent no. 213606]/Victor Taranenko, Antoni Świć, Dariusz Wołos, Georgiy Taranenko, Jakub Szabelski; author: Victor Taranenko, Antoni Świć, Dariusz Wołos, Georgiy Taranenko, Jakub Szabelski.-Patent no.; Patent application no.//Official Gazette of the Patent Office, 2013, No. 4, p. 851.Lathe tailstock [patent no. 214058]/Victor Taranenko, Antoni Świć, Dariusz Wołos, Georgiy Taranenko, Jakub Szabelski; author: Victor Taranenko, Antoni Świć, Dariusz Wołos, Gieorgij Taranenko, Jakub Szabelski.-Patent no.; Patent application no.//Official Gazette of the Patent Office, 2013, No. 6, p. 1426.1.

## Figures and Tables

**Figure 1 materials-14-04427-f001:**
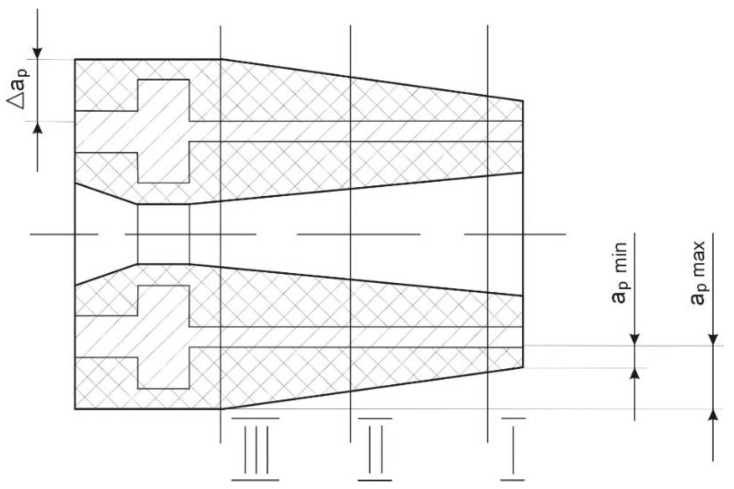
A semi-finished casing of an electric micro-machine obtained by pressing titanium powder.

**Figure 2 materials-14-04427-f002:**
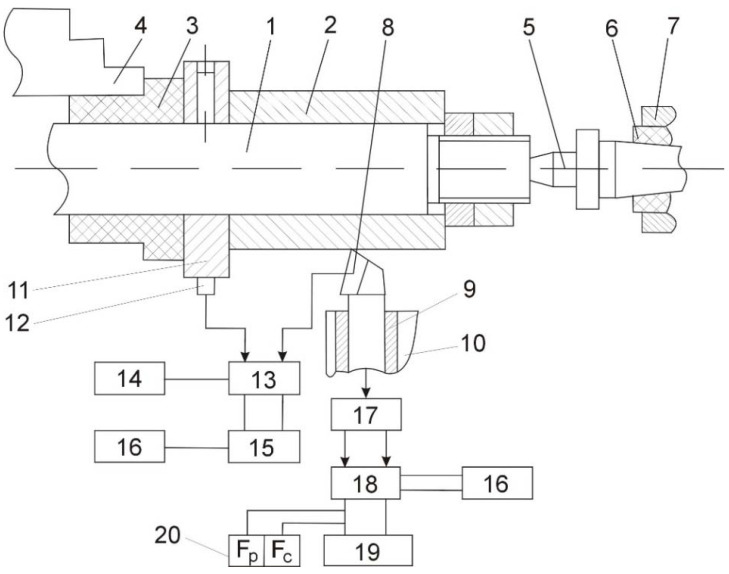
Schematic of the test stand: 1—work holding device, 2—workpiece, 3—dielectric sleeve, 4—machine tool holder, 5—rear center, 6—dielectric sleeve, 7—tailstock, 8—cutting edge, 9—tool holder, 10—tool post, 11—ring, 12—current measuring device, 13—vacuum switch, 14—electro-contact heating device, 15—voltmeter F4214, 16—ferroresonance voltage stabilizer, 17—dynamometer, 18—tensostaion UT4-1, 19—recording device N327/3, 20—measuring block.

**Figure 3 materials-14-04427-f003:**
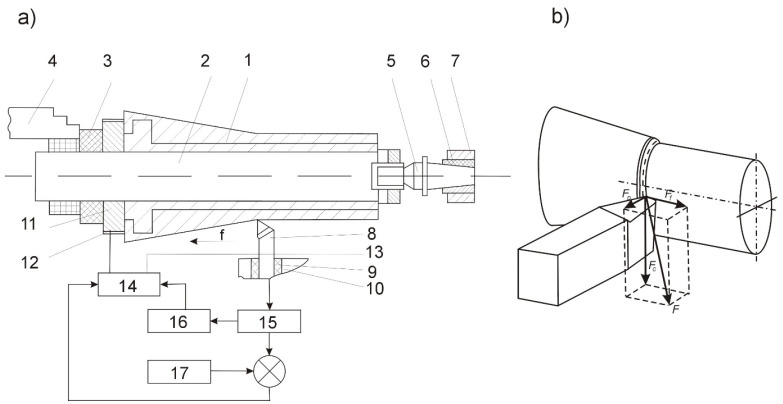
General schematic of machining process: (**a**) block diagram of the control system for electro-contact heating of the cutting zone: 1—workpiece, 2—work holding device, 3—dielectric sleeve, 4—tool holder, 5—rear center, 6—dielectric sleeve, 7—tailstock, 8—cutting tool, 9—dielectric holder, 10—clamp, 11—current feed ring, 12–13—contacts, 14—electro-contact heating device, 15—sensor, 16—threshold element, 17—program block, (**b**) decomposition of the main machining forces.

**Figure 4 materials-14-04427-f004:**
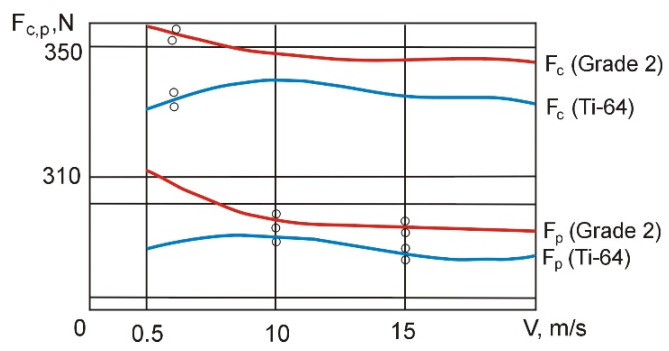
Experimental curves of changes in the *F_c_* and *F_p_* cutting force components as a function of the cutting speed during boring.

**Figure 5 materials-14-04427-f005:**
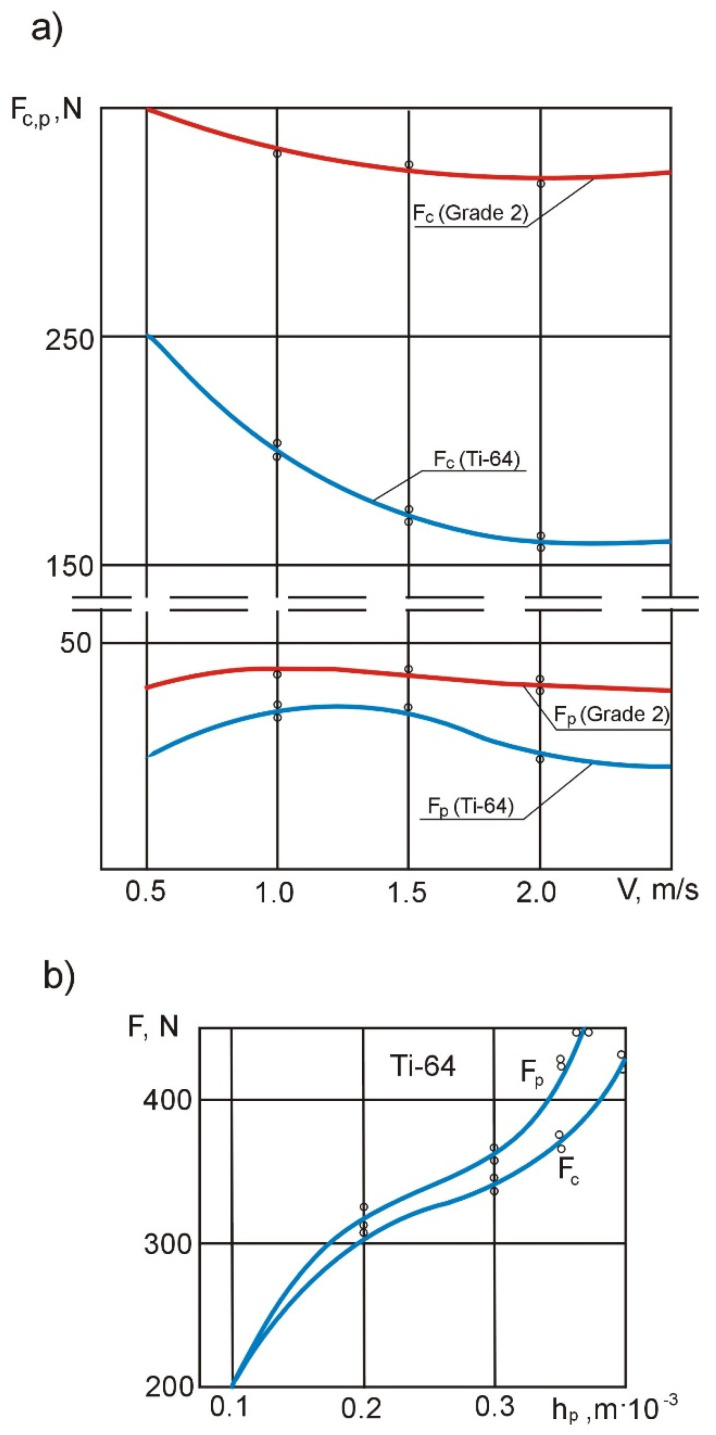
Curves of changes in *F_c_* and *F_p_* in the external turning operation as a function of the cutting speed (**a**) and wear *h_p_* under tension (**b**).

**Figure 6 materials-14-04427-f006:**
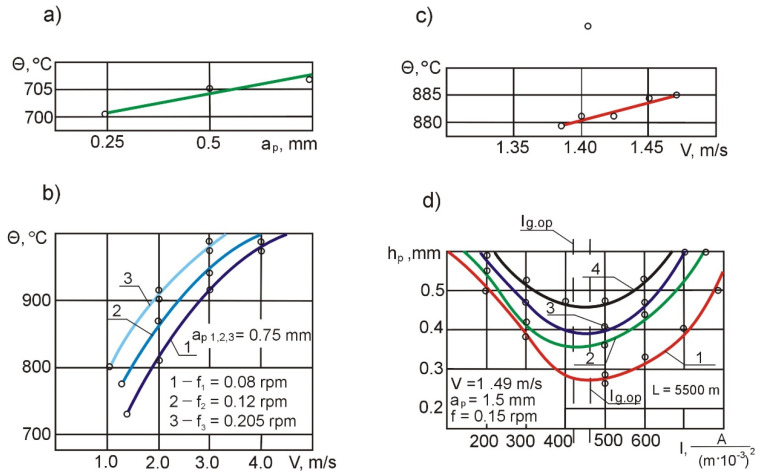
Effect of depth of cut on cutting zone temperature (**a**); effect of cutting speed on temperature change (**b**); cutting temperature changes during the machining of one part (**c**); flank wear under electro-contact heating: 1—*Grade 2*, turning; 2—*Grade 2*, boring; 3—*Ti-64*, turning; 4—*Ti-64*, boring (**d**).

**Figure 7 materials-14-04427-f007:**
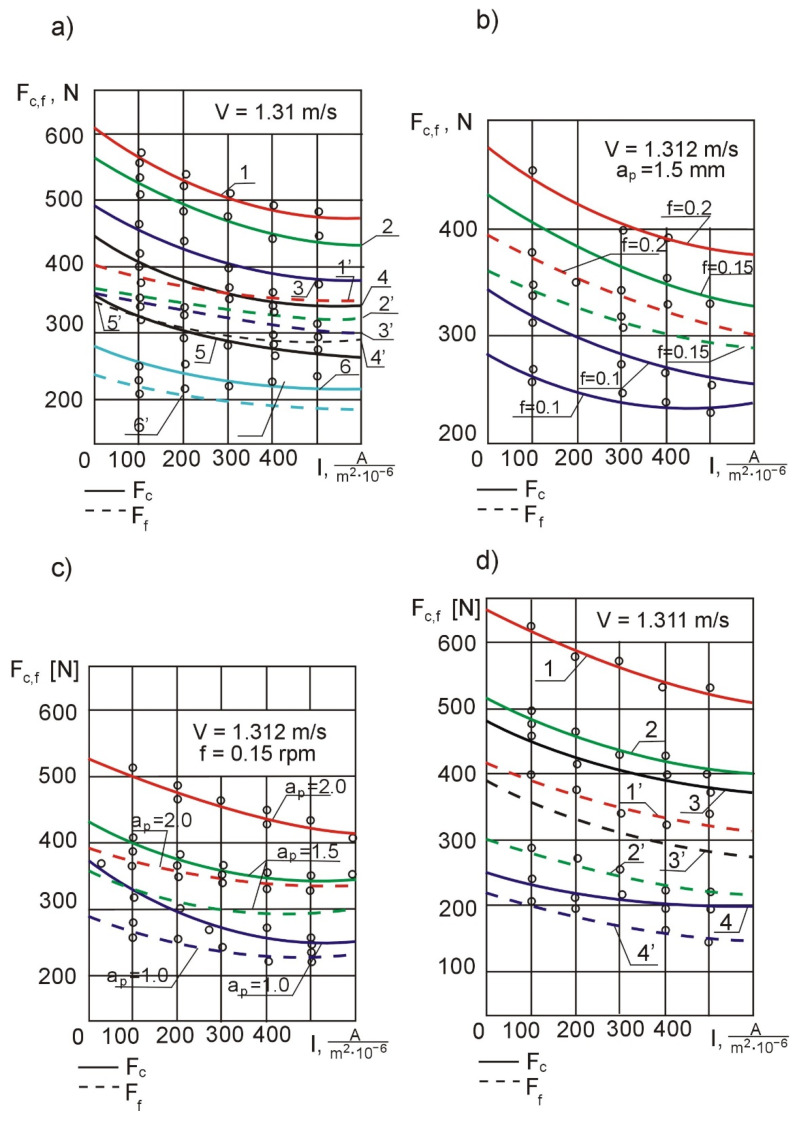
Effect of current density on cutting forces in the operation of turning *Grade 2* alloy: *f =* 0.2 ∙ 10^−3^ m/rev, *a_p_ =* 2 · 10^−3^ m(1–1’)*, f =* 0.15 · 10^−3^ m/rev, *a_p_ =* 2 *·* 10^−3^ m(2–2’), *f =* 0.2 · 10^−3^ m/rev, *a_p_ =* 1.5 · 10^−3^ m(3–3’), *f =* 0.1 ∙ 10^−3^ m/rev, *a_p_ =* 1.5 ∙ 10^−3^ m(4–4’), *f =* 0.1 ∙ 10^−3^ m/rev, *a_p_ =* 2 ∙ 10^−3^ m(5–5’), *f =* 0.1 ∙ 10^−3^ m/rev, *a_p_ =* 1.5 ∙ 10^−3^ m(6–6’)—(**a**); curves of changes in cutting forces generated during turning of *Ti-64* for different feed rates—(**b**); curves of changes in cutting forces generated during turning of *Ti-64* for different depths of cut—(**c**); cutting force curves for the *Ti-64* boring operation: *f =* 0.212 ∙ 10^−3^ m/rev, *a_p_ =* 1.5 ∙ 10^−3^ m(1–1’)*, f =* 0.15 ∙ 10^−3^ m/rev, *a_p_ =* 1 ∙ 10^−3^ m(2–2’), *f =* 0.15 ∙ 10^−3^ m/rev, *a_p_ =* 0.75 ∙ 10^−3^ m(3–3’), *a_p_ =* 1.5 ∙ 10^−3^ m(4–4’)—(**d**).

**Figure 8 materials-14-04427-f008:**
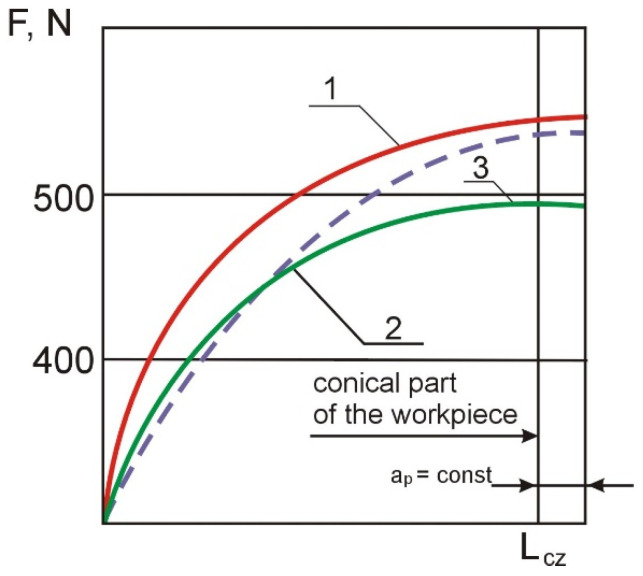
Curves of cutting forces vs. machining time: 1—without the use of electro-contact heating; 2—with the use of electro-contact heating; 3—with the use of the electro-contact heating control system.

**Figure 9 materials-14-04427-f009:**
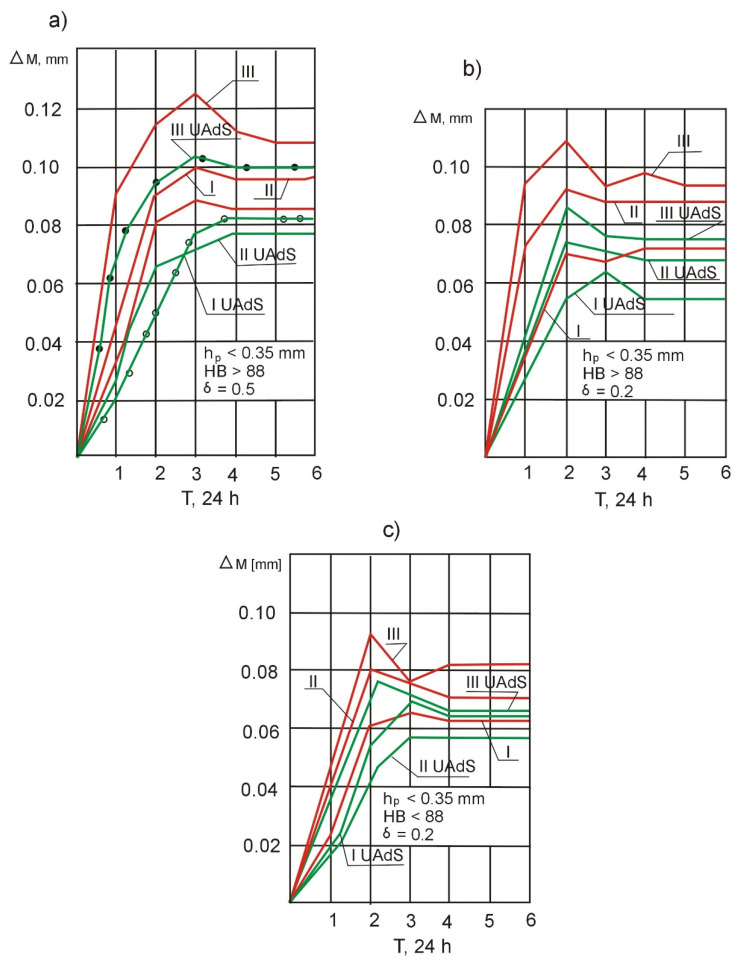
Changes in long-term dimensional stability of parts depending on the processing method used (with electro-contact heating control system—I and without the electro-contact heating control system—II) and the density of workpiece structure—III: (**a**) at maximum runout *δ* = 0.5 · 10^−3^ m; (**b**) at minimum runout *δ* = 0.2 · 10^−3^ m; and (**c**) at a minimum density of the *Ti-64* workpiece *HB* < 88.

**Table 1 materials-14-04427-t001:** Changing values of the cutting forces.

Cutting Forces *F*, [N]	Turning		Boring	
	Groove-Turning	Stabilization	Groove-Turning	Stabilization
*F_f_*	145	380	145	^_^
*F_p_*	^_^	^_^	^_^	480
*F* _c_	150	490	150	600

*F_f_*—feed (axial) cutting force, *F_p_*—thrust (radial) cutting force, *F_c_*—main (circumferential) cutting force.

**Table 2 materials-14-04427-t002:** Physicochemical properties of the cutting tool materials depending on the temperature.

Material	Physicomechanical Properties of Materials at *Q* = 20–1000 °C							
	*HV*		*σ_b_*		*ρ* · 10^−8^ Ω·m		*K_k_*	
	20°	1000°	20°	1000°	20°	1000°	20°	1000°
H30	1450	380	^−^	^_^	18.6	96	11	1
Grade 2	^_^	^_^	56	3	150	220	6	43
Ti-64	^_^	^_^	50	3	^_^	^_^	4	39

## Data Availability

All the data is available within the manuscript.

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
