# Peer review of "An Investigation into the Effect of Electro-Contact Heating in the Machining of Low-Rigidity Thin-Walled Micro-Machine Parts"

_materials, 2021, doi:10.3390/ma14164427_

Round 1
Reviewer 1 Report
Dear Authors,
Thank you for your well-written manuscript.
This manuscript is an original research and authors describe their innovative solution for measuring cutting forces and temperature at a specific speed, feed rate and depth of cut value. Also they developed a system to control electro-contact heating for their measurements. They clearly indicate in their introduction that not sufficient information is provided for certain technological aspects and that in general search of new methods is required. Even though they have a very short review, it contains all necessary information.
It is a very clear technical well-written manuscript. Conclusions are consistent and they address the work done described in the manuscript. The first author is also author of several patents.
Here will be following comments:
Lines 14-24: Please update the style of your writing in the manuscript, for example in your abstract instead of “we” in sentence “We also determined the effect of cutting parameters (depth of cut, feed rate and cutting speed) on temperature” -> use following description “The effect of cutting parameters (depth of cut, feed rate and cutting speed) on temperature has been also determined”.
Please indicate clearly the novelty in your abstract and obtained results.
Nomenclature is better to add at the end of the manuscript in appendix
Author Response
Dear Sir/Madame,
First of all, we would like to thank you very much for your time you spend on reviewing our paper and express our gratitude for your kind review. It was very supportive to read that you found our paper as well-written. However we would like to thank you for your remarks and suggestions too. They gave us opportunity to improve our paper. We enclose a table with detailed explanations and information about the changes that were made in the body of our manuscript.
Yours faithfully,
Authors

Reviewer 2 Report
It is known that the high quality of the machined parts in a short time is a research challenge for enhancing these parts' operating performance. This article investigates the dependency of cutting forces on cutting parameters and wear of flank, as well as changes in cutting forces induced by changes in heating current density and machining parameters during turning of thin-walled parts. The present paper is interesting and organized well. The methods and results are quite presented. The manuscript can be accepted in the current form.
Author Response
Dear Sir/Madame,
First of all, we would like to thank you very much for your time you spend on reviewing our paper and express our gratitude for your kind review. It was very supportive to read that you found our paper interesting as well-written. Thank you very much!
Yours faithfully,
Authors

Reviewer 3 Report
- For the multiple figures shown in this manuscript, the resolution, curve style, fonts of the legend are not ideal. The readers would not be able to understand the information easily at the first glance.
- It would be good to include several sentences of one paragraph to illustrate the goal of this study at the end of the Introduction section.
- In Fig. 2, the model names of the measurement tools (e.g. UT4-1, N327/3) are shown in the blocks. This is different from Fig. 3, in which only numbering is shown in the block. Please unify the style of block diagram. I prefer the style of Fig. 3 – explaining the meaning of each numbering underneath the figure.
- Please define the parameters shown in Table 2, HV and σb.
- It would be good to include a cartoon to explain the directions of Ff, Fp and Fc
- In Fig. 4, please show the unit of cutting speed
- In Fig.4 and 5, please define WT1-0, PTES-0, WTE-0. I cannot tell which curve is for Grade 2 and which curve is for Ti-64.
- What is “technological ultimate stress field”? Please define clearly what type of stress it is, tensile, shear, etc.
- In Fig. 6 , 7, 9, there is no label of (a) (b) (c) (d)
- “At the same time, tool life is doubled.” How do you reach such conclusion from the curve?
- Why the hp decreases with I first and then increases? Please explain why this phenomenon (optimal current density)
- The resolution of Fig. (9) is too low, hard to get the useful information.
Author Response
Dear Sir/Madame,
First of all, we would like to thank you very much for your time you spend on reviewing our paper and express our gratitude for your kind review. Moreover we would like to thank you for your remarks and suggestions. They undoubtedly gave us opportunity to improve our paper. We enclose a table where you can find detailed explanations and information about the changes that were made in the body of our manuscript.
Yours faithfully,
Authors

This manuscript is a resubmission of an earlier submission. The following is a list of the peer review reports and author responses from that submission.